# Bioabsorbable Carboxymethyl Starch–Calcium Ionic Assembly Powder as a Hemostatic Agent

**DOI:** 10.3390/polym14183909

**Published:** 2022-09-19

**Authors:** Young-Gwang Ko, Byeong Nam Kim, Eun Jin Kim, Ho Yun Chung, Seong Yong Park, Young-Jin Kim, Oh Hyeong Kwon

**Affiliations:** 1Department of Polymer Science and Engineering, Kumoh National Institute of Technology, Gumi 39177, Korea; 2Theracion Biomedical Co., Ltd., Seongnam 13201, Korea; 3Department of Plastic and Reconstructive Surgery, CMRI, Kyungpook National University School of Medicine, Daegu 41944, Korea; 4Department of Thoracic and Cardiovascular Surgery, Samsung Medical Center, Sungkyunkwan University School of Medicine, Seoul 06351, Korea; 5Department of Advanced Materials and Chemical Engineering, Daegu Catholic University, Gyeongsan 38430, Korea

**Keywords:** powder, carboxymethyl starch, calcium, hemostatic agent

## Abstract

In contrast to hemostatic fabrics, foams, and gels, hemostatic spray powders may be conveniently applied on narrow and complex bleeding sites. However, powdered hemostatic agents are easily desorbed from the bleeding surface because of blood flow, which seriously decreases their hemostatic function. In this study, the hemostatic performance of a bioabsorbable powder with decreased desorption was investigated. The proposed hemostatic powder (OOZFIX^TM^) is an ionic assembly of carboxymethyl starch and calcium. The microstructure and chemical properties of the hemostatic powder were analyzed. The hemostatic performance (blood absorption, blood absorption rate, and coagulation time), thromboelastography (TEG), rheology, adhesion force, and C3a complement activation of the OOZFIX^TM^ were evaluated and compared with those of the carboxymethyl starch-based commercial hemostatic powder (Arista^TM^ AH). The in vivo rat hepatic hemorrhage model for hemostasis time and bioabsorption of the OOZFIX^TM^ showed quick biodegradation (<3 weeks) and a significantly improved hemostasis rate (78 ± 17 s) compared to that of Arista^TM^ AH (182 ± 11) because of the reduced desorption. The bioabsorbable hemostatic powder OOZFIX^TM^ is expected to be a promising hemostatic agent for precise medical surgical treatments.

## 1. Introduction

In recent years, the frequency of surgical operations has considerably increased owing to developments in medical technology and operation convenience [1]. Bleeding from local wounds formed during simple surgical procedures is sufficiently controlled by bodily hemostasis. The amount of blood in an average adult is approximately 6–7% of body weight, and blood loss presents a risk to survival only when it accounts for more than 10% of the total blood volume [2]. When the body is wounded, epithelialization proceeds through hemostasis and the inflammatory phase within the first day, followed by the proliferation of epithelial cells for 3 weeks [3,4]. Hemostatic agents are applied at the earliest stage of medical treatment to wound sites to prevent bleeding and blood loss [5,6,7,8,9,10].

There are multiple methods to induce hemostasis: mechanical, chemical, cooling, and heating. The mechanical methods include direct pressure with gauze, sutures, clamps, clips, and other devices. Chemical methods include astringents, blood coagulants, and local hemostatic agents. The cooling and heating methods include ice and electric heat-generating devices. Recently, hemostatic agents based on positively charged polymer materials (blood components have a negative charge) have been developed [11,12,13,14]. Oxidatively regenerated cellulose (Surgicel^®^, Ethicon, Somerville, NJ, USA) that promotes hemostasis may be used in packings, gauzes, or compressed sponges; however, it can cause pressure and pain during packing. Therefore, absorbent packings have been developed to compensate for this disadvantage. Representative examples include gelatin sponge (Gelfoam^®^, Pfizer, New York, NY, USA), bioabsorbable polyurethane foam (Nasopore^®^, Polyganics, Netherlands), gelatin/thrombin mixture (Floseal^®^, Baxter, Deerfield, IL, USA), hyaluronic acid sponge (MeroGel^®^/Meropack^®^, Medtronic, Jacksonville, FL, USA), carboxymethylcellulose mesh (Sinu-Knit, ArthroCare^®^, Glenfield, UK), and absorbable collagen sponge (Avitene^®^, Davol, Cranston, RI, USA) [4,5,11]. However, commercial absorptive packings are inconvenient for use in abdominal surgery for complex and narrow hemorrhage sites. Therefore, patient-friendly hemostatic agents with an excellent hemostatic effect for application in complex and narrow sites need to be developed.

In this study, an effective hemostatic agent addressing the above problems is proposed. A powder-type highly absorbent adhesive hemostatic agent containing blood coagulation factors (calcium chloride) and with carboxymethyl starch was designed. Powder-type hemostatic agents are easy to apply even in narrow and complicated areas and generally, do not cause intense foreign body sensations [15,16]. In addition, the proposed hemostatic powder was intended to maintain good hemostasis performance without desorption because of carboxymethyl starch and calcium ionic assembly network. Fine particles have a high specific surface area and coating ability; thus, they are widely used in the medical field [17,18,19,20]. In particular, microparticles have very small diameters and a large specific surface area enabling good blood plasma absorption, thereby maximizing the effect of blood coagulation factors.

Here, we demonstrated blood plasma absorption and blood coagulations of bioabsorbable hemostatic powder composed of carboxymethyl starch and calcium ions. The developed hemostatic powder is highly promising for use in clinical applications and surgery.

## 2. Materials and Methods

### 2.1. Fabrication of Hemostatic Powder

Carboxymethyl starch (50 wt%; JRS PHARMA GmbH, Germany) was dissolved and stirred in a solution of calcium chloride (0.08 wt%; CaCl_2,_ Daejung, Siheung, Korea) in ethanol (99%, Daejung, Siheung, Korea) for 2 h. The obtained solution containing precipitate was filtered using a filter paper to eliminate unreacted calcium chloride, which was then rinsed with ethanol three times. Subsequently, the filtered solution was dried in an oven (40 °C for 24 h) and pulverized by cryogenic grinding. The ground hemostatic particles were then sieved using a sieve tray (nominal aperture 125 μm, Daihan Scientific, Seoul, Korea) to obtain a homogeneous powder. The powdered sample was sterilized under UV light (254 nm) and kept in a sealed pack for further use.

### 2.2. Microstructure and Chemical Analysis

The morphology of the hemostatic powder was investigated using scanning electron microscopy (SEM, JSM-6380, JEOL Ltd., Tokyo, Japan) with an acceleration voltage range of 11–15 kV after sputter-coating with platinum. The average diameter (*D*) of the polygonal powder particles was determined as the equivalent area diameter (*D*^2^ = 4 *A*/π, where *A* is the cross-sectional particle area) using image analysis software (I-solution Lite, IMT i-solution, Seoul, Korea) and a particle size analyzer (Shimadzu SALD-2300, WingSALD II, Version 3.3.0, Kyoto, Japan). The powder particle size was measured in triplicate for each specimen.

The chemical compositions of hemostatic powders were investigated using an attenuated total reflection (ATR)-Fourier transform infrared (FTIR) spectroscopy (VERTEX 80V, Bruker, Ettlingen, Germany) with a germanium ATR crystal. All spectra were recorded in absorption mode with a 4.0 cm^−1^ scan interval in the scanning range of 4000–800 cm^−1^.

### 2.3. In Vitro Blood Absorption and Coagulation Time (Lee–White Method)

Quantitative analysis of the blood absorption amount, blood absorption rate, and coagulation ability is essential for evaluating the hemostatic performance. The absorption and coagulation characteristics of the blood treated with the hemostatic powder were analyzed using whole blood (dog, 13% anticoagulant, Korea Animal Blood Bank, Seoul, Korea). CaCl_2_ (0.0125 M) was dissolved in the blood prior to the experiment to avoid any anticoagulant effects. In brief, sample powder (0.1 g) was placed in a watch glass, and then the blood was dropped onto it until saturation. At the moment of saturation, the blood volume and rate (elapsed time) were recorded. The blood absorption was calculated using the following equation:*Blood absorption* (mL/g) = *absorbed blood amount* (mL)/*sample weight* (g)

Blood coagulation time was measured using the Lee–White method [16,21]. A glass vial was placed in a 37 °C water bath for 10 min. Then, blood (3 mL) and hemostatic powder (0.1 g) were added into the vial, and the vial was tilted every 20 s. The time until blood no longer flowed after contacting the hemostatic agent was recorded.

### 2.4. TEG

The blood-clotting activity of human blood exposed to the hemostatic powder was analyzed using TEG assays. The TEG machine (TEG Hemostasis Analyzer 5000; Haemoscope Corporation, Niles, IL, USA) was calibrated before use according to quality control standards. TEG assays were performed within 2 h of sample collection, in accordance with the manufacturer’s instructions. To study the effect of powder dosage on coagulation function, 5 mL of whole blood were collected in a tube containing citrate (3.2%). The specimen (0.01 g), whole blood (340 μL, dog, Korea Animal Blood Bank, Sokcho, Korea), and CaCl_2_ (40 μL) were placed in a TEG cup and analyzed using a plastic TEG instrument (39 °C). The reaction time (*R*, min), clotting time (*K*, min), angle (α, °), and maximum amplitude (MA, mm) were recorded. Whole blood (5.4 mL) was collected from other 10 healthy blood donors in 2 tubes (2.7 mL per tube) containing citrate (3.2%); an additional 2 mL of whole blood was collected in a tube containing EDTA-k2. Using these samples, conventional coagulation tests, TEG assays (with 20 mg of each powder), and blood cell counts were performed.

### 2.5. Rheology

The storage modulus (*G*’), loss modulus (*G*″), and delta (δ) of the hemostatic agents were determined using a rheometer (ARES, TA instruments, USA, 25 °C, gap: 1 mm, plate diameter: 10 mm, oscillation strain: 0.01–100%, frequency: 10 Hz). The specimens were analyzed in a gel state obtained by adding 7.5% water to the powder.

### 2.6. Adhesion Ability

The adhesion force was measured by the falling force between the tissue and the coagulated powder sample. Briefly, a piece of pig skin was attached to the upper surface (40 mm in diameter parallel Peltier steel plate, 37 °C) of the rheometer (ARES, TA instruments, New Castle, DE, USA). The powder (0.2 g) was placed on the bottom plate of the device, mixed with 1 mL of purified water, and allowed to swell. The swollen sample was applied with an upward force to the pig skin attached to the top of the rheometer.

### 2.7. Cytotoxicity

Elution test of bioabsorbable hemostatic powder was performed using L929 fibroblasts (connective mouse tissue) to investigate cytotoxicity (ISO 10993-5: 2009, Biological Evaluation of Medical Devices). The test substance and control substance were eluted with minimum essential medium (MEM) culture solution containing 10% fetal bovine serum (FBS), 1% penicillin streptomycin in an incubator (37 °C, 5% CO_2_) for 24 h. Test materials eluate (test group) in L-929 cells were high-density polyethylene (HDPE) film (Hatano Research Institute, Japan) as a negative control material (negative control group), 0.25% zinc dibuthyldithiocarbamate (ZDBC) polyurethane film (Hatano Research Institute, Japan) as a positive control material (positive control group), blank test solution (solvent control group), and OOZFIX^TM^ powder. Samples were incubated in an incubator for 48 h.

For qualitative evaluation, it was determined under ISO 10993-5 regulation. Eluate of the test substance had no cytotoxicity if it was grade 2 or less. Quantitative evaluation was carried out after qualitative evaluation, and Trypsin-EDTA was treated in each cell culture well and incubated for about 5 min in an incubator. Relative cell count (RCC) was calculated as the following equation.
$$\mathrm{RCC}\;(\mathrm{Relative}\;\mathrm{Cell}\;\mathrm{Count})\;\%=\frac{Number\;of\;cells\;in\;the\;test\;group\;\;\left(\mathrm{cell}/\mathrm{mL}\right)}{Number\;of\;cells\;in\;the\;solvent\;control\;group\left(\mathrm{cell}/\mathrm{mL}\right)}\times 100$$

If the viability decreased to less than 70% (<70%) of the blank group, it was judged to be potentially cytotoxic.

### 2.8. C3a Complement Activation Assay

An enzymatic immunoassay for the quantification of C3a fragments present in the human blood after the interaction with hemostatic powders was performed using the C3a PlusEIA kit from MicroVue (Quidel, San Diego, CA, USA) and immunosorbent assay (PHOmo, Autobio, Zhengzhou, China). The powder samples were incubated at 37 °C for 1 h and subsequently processed according to the manufacturer’s instructions. The absorbance was measured at 450 nm using a microplate reader (Anthos HT III, type 12,600, Anthos Mikrosysteme GmbH, Krefeld, Germany). The concentration of C3a was expressed in ng/mL and as a percentage of the activation in the control, which was incubated and treated using the same procedure. Zymosan (Sigma Aldrich, St Louis, MO, USA) was used as the positive control. Measurements were performed in duplicate.

### 2.9. In Vivo Hemostasis and Biodegradation

Outbred Sprague–Dawley rats (male, 230–280 g, 8 weeks old, Hyochang Science, Daegu, Korea) were used as experimental models for evaluating in vivo hemostasis and biodegradation. All animal experiments were reviewed and approved by the Institutional Animal Care and Use Ethics Committee of the Kyungpook National University (No.: 2020-0105, Date: 24 October 2020). The experimental procedures were approved by the Animal Care Committee and performed as follows. The rats were anesthetized using a 4:1 volume ratio of Zoletil (Virbac, France) to Rompun (Bayer, Germany). Following stabilization of the rats, their livers beside the hilus were opened, and bleeding was induced using punch biopsy (6 mm) with a cross (+) shape (1 × 1 cm). The bleeding site was covered with cotton gauze, and after the removal of the gauze, the powder samples (0.1 g) were applied. Thereafter, a 50-gram weight block was placed onto the site for 60 s. The site was observed every 15 s and rinsed with a saline solution.

The absorbable hemostatic powder does not require the removal of the blood-clot complex material. We performed in vivo experiments to evaluate the biodegradability of OOZFIX^TM^ powder. The powder (0.1 g) was injected intradermally into the backs of the rats (*n* = 16). The biodegradability of the powder was evaluated at weeks 1, 2, 3, and 4.

### 2.10. Statistical Analysis

All data are presented as mean ± standard deviation. Statistical analyses were performed using KyPlot software version 6.0 (KyensLab, Inc., Tokyo, Japan). Significance levels were calculated using parametric Student’s *t*-tests and one-way ANOVA with a Tukey’s post hoc test. Statistical significance was set to *p* < 0.05.

## 3. Results and Discussion

### 3.1. Microstructure of Hemostatic Powders

The particle size and surface morphology are the key parameters for assessing the blood absorption and clotting behavior of hemostatic powders with the same base. This is because the specific surface area of the particles and the volume of voids allowing quick blood absorption are proportional to the particle dimensions and surface structure; thus, a high specific area leads to rapid hemostasis. The OOZFIX^TM^ construct was prepared via ionic assembly of carboxymethyl starch with CaCl_2_. The dried construct was cryogenically ground and sieved to obtain a homogeneous powder. As shown in Figure 1, the particle sizes of the OOZFIX^TM^ and Arista^TM^ AH powders are similar, with the mean diameter ranging from 30 to 150 μm. The OOZFIX^TM^ powder (mean diameter: 64.4 μm) consists of spheroidal particles with a relatively smooth surface. The Arista^TM^ AH (mean diameter: 65.5 μm) powder particles are spherical but have a wrinkled surface. Singh reported that irregularities in non-spherical particles of hemostatic powders contained more voids for blood absorption and clot formation [22,23]. Although the microstructure of both powders is not identical, the mean diameter of globular powders has a similar distribution with negligible voids. Therefore, the results suggest that OOZFIX^TM^ and Arista^TM^ AH powder can absorb blood plasma in a similar time.

### 3.2. Chemical Analysis

The functional groups of the hemostatic agent were confirmed using ATR-FTIR analysis (Figure 2). The base material of both the OOZFIX^TM^ and Arista^TM^ AH powders is carboxymethyl starch. The characteristic spectra of carboxymethyl starch show broad peaks at 3430 and 2930 cm^−1^, corresponding to the stretching vibrations of –OH and –CH_2_, respectively. The peaks at 1650 and 1365 cm^−1^ were assigned to the –OH bending vibration. The peaks at 1000–1200 cm^−1^ were ascribed to the stretching of the –C–O–C– linkage in the polysaccharide backbone chain. The ATR-FTIR spectrum of the OOZFIX^TM^ powder shows peaks at 1612 and 1425 cm^−1^, which are not observed in the spectrum of the Arista^TM^ AH powder. We attributed these peaks in OOZFIX^TM^ to the –COO^−^ asymmetric and symmetric stretching vibrations, respectively, caused by the ionic assembly with Ca^2+^ [10].

### 3.3. In Vitro Blood Absorption and Coagulation Time

During in vitro evaluation, the amount and rate of blood absorption until maximum absorption are measured by contacting the blood with a hemostatic powder. As shown in Figure 3A, OOZFIX^TM^ and Arista^TM^ AH powders exhibit similar blood absorption amounts and both show a Grade 1 blood absorption rate (less than 5 s). The amount and rate of blood absorption are proportional to the source materials and specific surface area [3,4,5]. As shown in Figure 1, the powders have the same base material and similar particle sizes. Therefore, the amount and rate of blood absorption do not differ significantly.

Shorter coagulation time indicates the effective activation of blood coagulation factors and stronger adhesion of platelets. As shown in Figure 3B, the blood coagulation time in the OOZFIX^TM^ group is significantly lower than that in the Arista^TM^ AH group. Even though the blood absorption amounts of both hemostatic powders are similar, the OOZFIX^TM^ group shows faster blood coagulation than the Arista^TM^ AH group. After blood absorption, OOZFIX^TM^ maintained its network structure because of ionic assembly with Ca^2+^ that promoted blood coagulation. Therefore, we concluded that Ca^2+^ on the carboxymethyl starch network structure facilitated hydrophilicity and blood coagulation with gelation at later steps of hemostasis.

### 3.4. TEG

The TEG parameters (*R*, *K*, α, and MA) of the OOZFIX^TM^ and Arista^TM^ AH powders are summarized in Table 1. The fibrin and clot formation profiles (TEG tracings) of the blood samples treated with hemostatic powders are shown in Figure 4. The *R* and *K* values for the OOZFIX^TM^ group (*R* = 3.6, *K* = 0.8) are lower than those for the Arista^TM^ AH group (*R* = 4.6, *K* = 1.6). Clotting parameters α and MA for the OOZFIX^TM^ group (α = 82.4, MA = 83.1) are much higher than those for the Arista^TM^ AH group (α = 59.2, MA = 75.1). The TEG data support the results of the blood coagulation time test (Figure 3B), indicating lower initial fibrin coagulation and clot formation times in the OOZFIX^TM^ group. In addition, the maximum clot strength (MA) in the OOZFIX^TM^ group is higher than that in the Arista^TM^ AH group.

### 3.5. Rheology

Viscoelasticity of hemostatic powders reflects the physical strength of the gelated powder on wound skin site after blood absorption and hemostasis. As shown in Figure 5, the OOZFIX^TM^ and Arista^TM^ AH gels exhibit the rubbery plateau region storage (*G’*, elasticity) of 3000 and 1000 Pa, and loss moduli (*G’’*, viscosity) of 500 and 100 Pa, respectively. The viscoelastic gel properties on plateau region and maximum delta values (δ, damping behavior) of the OOZFIX^TM^ gels maintain higher values than those of the Arista^TM^ AH gels. This means that the OOZFIX^TM^ gels are expected to maintain on the wound skin despite increasing strain after blood absorption and coagulation.

### 3.6. Adhesion

Bioabsorbable hemostatic powders have been widely adopted for the convenience of their use in narrow and complicated sites, particularly during laparoscopy. However, their efficacy is limited because of partial desorption at the bleeding site, which is not observed with other types (fabric, sponge, gel) of hemostatic agents [16,17]. A tissue-adhesive hemostatic powder would solve this issue and show fair hemostatic ability without serious loss of hemostatic powder. As shown in Figure 6, the adhesive strength of the OOZFIX^TM^ group is −0.45 N, which is twice that of the Arista^TM^ AH group. The OOZFIX^TM^ powder is a carboxymethyl starch-calcium ion assembly [15], and its hydrophilicity increases with hydrogelation through blood plasma absorption, thus enabling strong adhesion of the OOZFIX^TM^ gels.

### 3.7. Cytotoxicity for a Bioabsorbable Hemostatic Agent

Quantitative cytotoxicity of powder-type hemostatic agents was evaluated by counting live cells cultured in the eluate of the sample (Figure 7). Compared with the positive control, the negative control and OOZFIX^TM^ showed cell viability of 96.64% and 94.62%, respectively. In addition, qualitative analysis was performed through microscopic observation of a cell monolayer. Compared with the positive control, the negative control and OOZFIX^TM^ were hardly observed to separate the extracellular matrix and inhibit the growth of cells, and showed grades 0 and 1, respectively. Although foreign body reactions of natural carbohydrates are rare, investigating cytotoxicity of a new powder product made of polysaccharides (OOZFIX^TM^) is important. In both quantitative and qualitative results, the OOZFIX^TM^ powder-type hemostatic agent was judged to be safe to apply as a medical material.

### 3.8. C3a Complement Activation

Complement system activation is a major parameter for evaluating the inflammatory reaction to hemostatic agents. The complement system is critical for inducing the immunogenic response to bacteria, viruses, and other foreign bodies. Activation of this system can proceed through three possible pathways, which all involve multistep protein cleavage, but differ by the nature of the triggering element. These three enzymatic cascades have a common enzyme, C3 convertase, which cleaves the C3 protein in anaphylatoxin C3a. The level of C3a detected in the blood incubated with the OOZFIX^TM^ and Arista^TM^ AH powders reflects the inflammation reaction. As shown in Figure 8, the Arista^TM^ AH powder exhibits significant activation.

### 3.9. In Vivo Hemostasis Time

Hemostatic evaluation through animal model experiments is the most important method for analyzing the performance of hemostatic agents. As shown in Figure 9, the average hemostasis times in groups treated with the OOZFIX^TM^ and Arista^TM^ AH powders are 78 ± 17 and 182 ± 11 s, respectively. The OOZFIX^TM^ group showed a significant reduction in hemostasis time. Several hemostatic powders contain both blood absorption materials and blood-clotting reagents, such as thrombin and CaCl_2_, to facilitate clotting [12,15]. Therefore, the excellent hemostatic performance of the OOZFIX^TM^ powder was attributed to the synergistic effects of carboxymethyl starch and CaCl_2_.

### 3.10. In Vivo Biodegradation

OOZFIX^TM^ hemostatic powder was designed as a biodegradable agent for hemostasis without removal [24]. Thus, the biodegradation behavior of OOZFIX^TM^ powder was investigated in an in vivo environment (intradermal injection into rat scapula) by observing the morphology after a set period. As shown in Figure 10, the OOZFIX^TM^ powder remained after 1 week, while most of the powder was desorbed with partial debris after 2 weeks. No powder traces were observed in the intradermal areas after 3 and 4 weeks. The biodegradation rates of hemostatic agents vary depending on the base material and type (fabric, foam, gel, or powder). The OOZFIX^TM^ powder is based on carboxymethyl starch, which is a plant-based bioabsorbable material. Simultaneously, because of the large specific surface area, the hemostatic powder is quickly biodegraded after inducing hemostasis, thus relieving the uncomfortable feelings after laparoscopic surgery in narrow and complicated areas.

## 4. Conclusions

A hemostatic powder (OOZFIX^TM^) composed of carboxymethyl starch and calcium ions was fabricated via ionic assembly, drying, cryogrinding, and sieving. The OOZFIX^TM^ powder consisted of spheroidal particles. The in vitro blood absorption amount and rate in the OOZFIX^TM^ group were similar to those in the Arista^TM^ AH group. In contrast, the blood coagulation rate and clot strength in the OOZFIX^TM^ group were significantly better than in the Arista^TM^ AH group. The OOZFIX^TM^ group showed lower levels of C3a complement activity than the Arista^TM^ AH group. In vivo rat hepatic hemorrhage model experiment showed remarkable hemostatic ability in the OOZFIX^TM^ group owing to reduced desorption. The biocompatibility and hemostatic properties of the carboxymethyl starch–calcium powder (OOZFIX^TM^) developed in this study were superior to those of commercialized products. Moreover, incorporating active blood coagulants, such as thrombin, in powder-type hemostatic agents is a challengeable agenda for future works. In order to expand the application area of powder-type hemostatic agents, functional hemostatic properties are required based on this study.

## Figures and Tables

**Figure 1 polymers-14-03909-f001:**
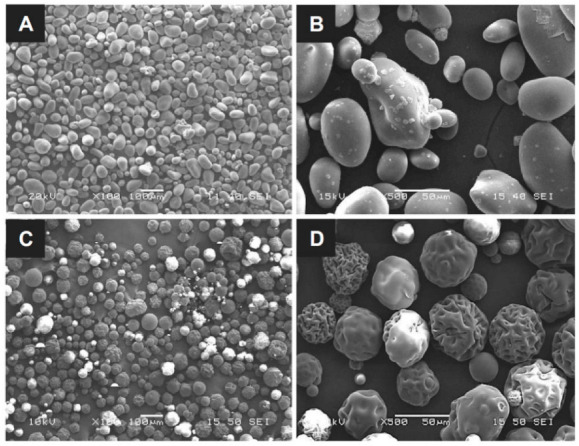
Micrographs of the (**A**,**B**) OOZFIX^TM^ and (**C**,**D**) Arista^TM^ AH hemostatic powders. Scale bars in (**A**,**C**) are 100 μm, and in (**B**,**D**) are 50 μm.

**Figure 2 polymers-14-03909-f002:**
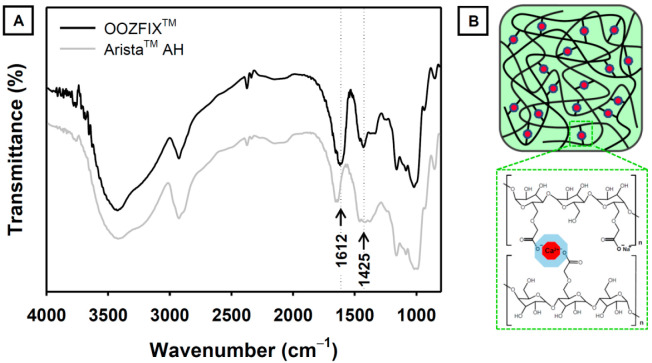
(**A**) Attenuated total reflection-Fourier transform infrared spectra of the OOZFIX^TM^ and Arista^TM^ AH hemostatic powders. The new peak in OOZFIX^TM^ at 1612 cm^−1^ was attributed to stretching vibrations of the carboxymethyl group in the ionic assembly with calcium. (**B**) Schematic of ionic assembly between carboxymethyl starch branches and calcium ions.

**Figure 3 polymers-14-03909-f003:**
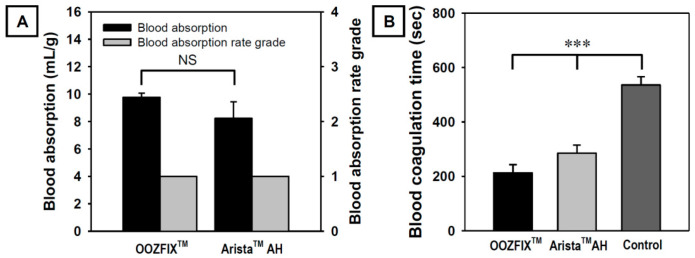
(**A**) In vitro blood absorption and blood absorption rate grade (*n* = 3) and (**B**) in vitro blood coagulation time (Lee–White method, *n* = 5) of the OOZFIX^TM^ and Arista^TM^ AH hemostatic powders. Grade 1: <5 s, Grade 2: 5–10 s, Grade 3: 10–30 s, Grade 4: >40 s (NS: not significant, *** *p* < 0.001).

**Figure 4 polymers-14-03909-f004:**
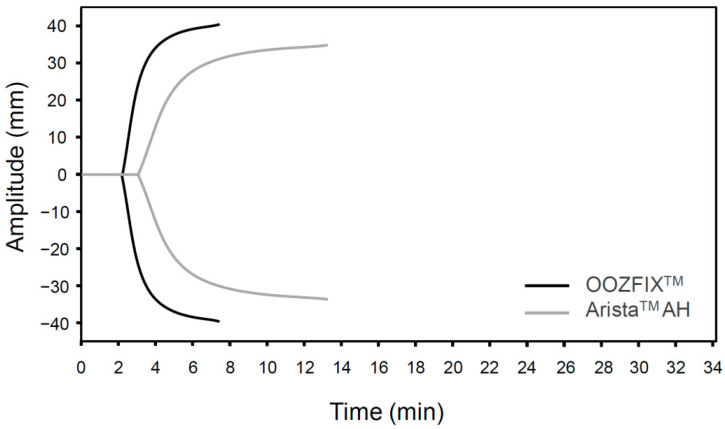
TEG of hemostatic powders. Representative data are selected from three repeated measurements.

**Figure 5 polymers-14-03909-f005:**
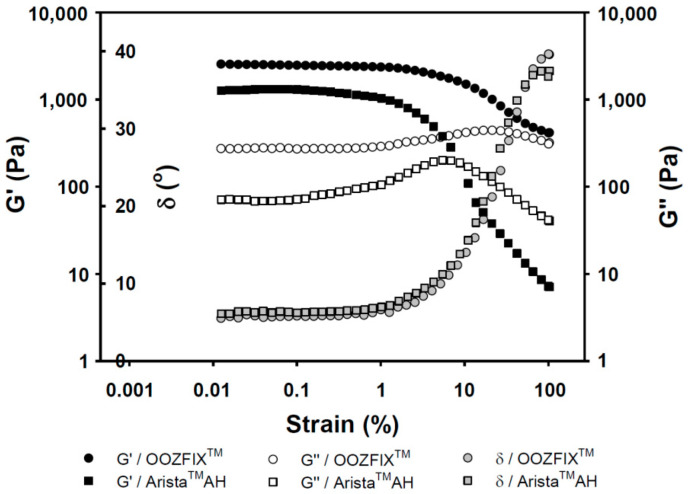
Rheological gelation properties (*G*’: storage modulus, *G*″: loss modulus, and δ: damping, tangent 45° = 1) against strain for the hemostatic agents after water absorption (gel state containing 7.5% water).

**Figure 6 polymers-14-03909-f006:**
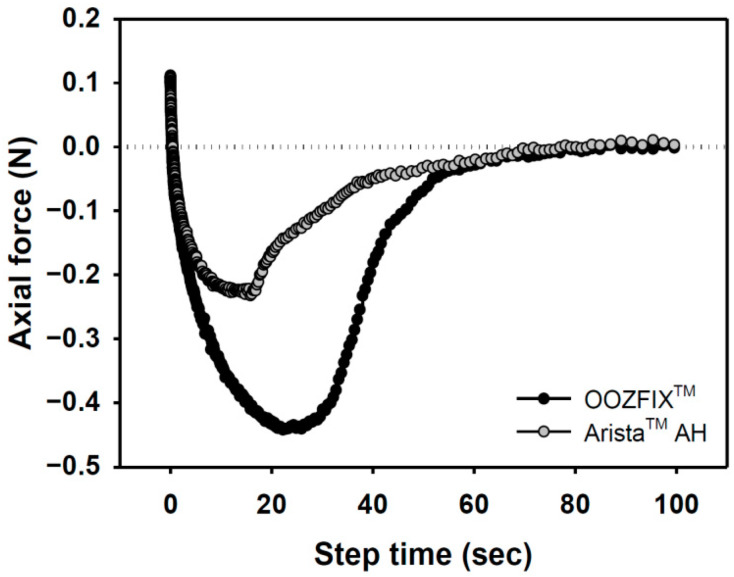
Adhesion force of hemostatic agents on pig skin vs. step time after water absorption (gel state containing 7.5% water).

**Figure 7 polymers-14-03909-f007:**
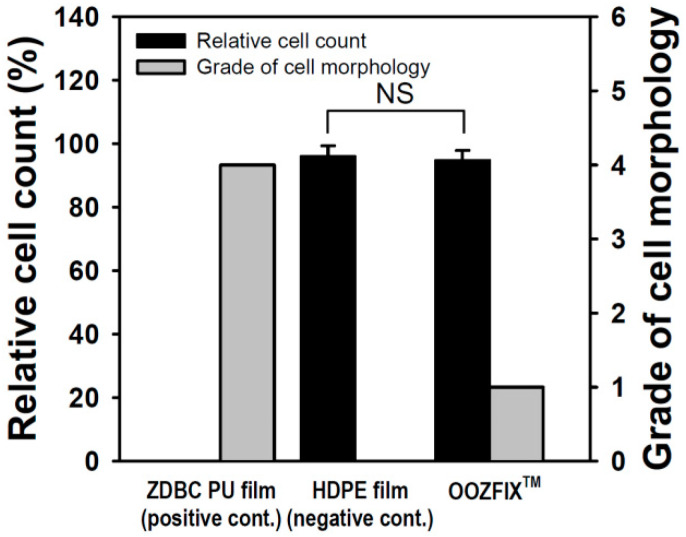
Cytotoxicity evaluation of the hemostatic powder (n = 3, NS means not significant).

**Figure 8 polymers-14-03909-f008:**
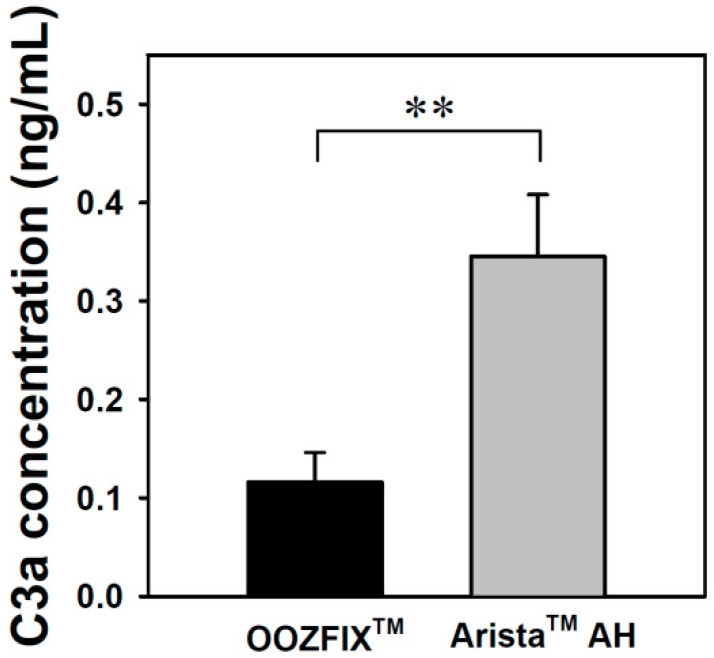
C3a complement system activation assay of the hemostatic powders in human blood (n = 3, ** *p* < 0.01).

**Figure 9 polymers-14-03909-f009:**
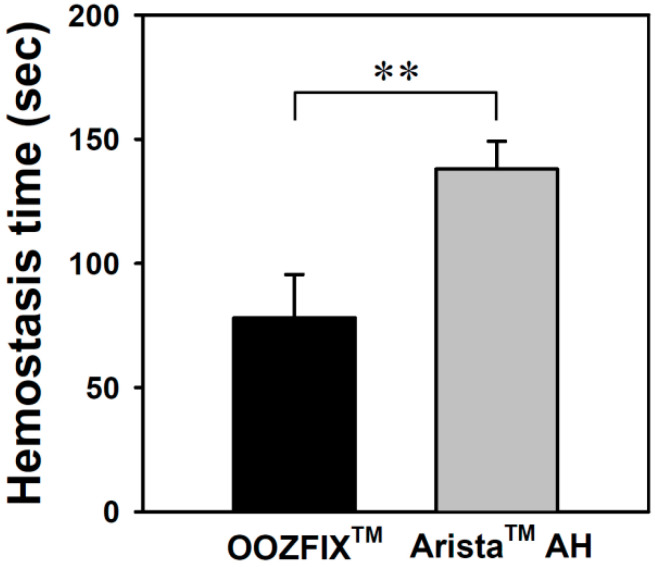
In vivo hemostasis time of the hemostatic powders in rat hepatic hemorrhage model (n = 5, ** *p* < 0.01).

**Figure 10 polymers-14-03909-f010:**
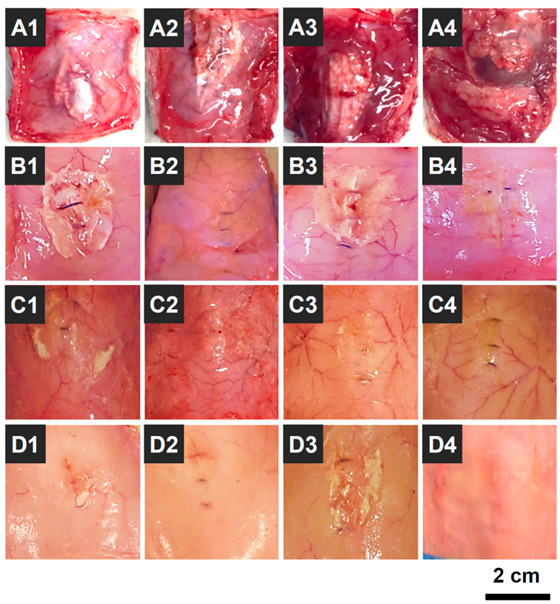
Photographs of in vivo biodegradation of the OOZFIX^TM^ hemostatic powder (0.2 g) that was intradermally injected into the rat scapula (*n* = 16); (**A1**–**A4**) 1, (**B1**–**B4**) 2, (**C1**–**C4**) 3, and (**D1**–**D4**) 4 weeks after the injection.

**Table 1 polymers-14-03909-t001:** Blood coagulation parameters: thromboelastography (TEG) data for hemostatic powders.

	^1^ Reaction Time (min)	^2^ Clotting Time (min)	^3^ Alpha Angle(°)	^4^ Maximum Amplitude (mm)
OOZFIX^TM^	3.6	0.8	82.4	83.1
Arista^TM^ AH	4.6	1.6	59.2	75.1

^1^*R*: time of initial fibrin formation (defined by an amplitude of 2 mm); ^2^*K*: time of initial clot formation as thrombin and platelet interaction (from the end of *R* time until the amplitude of 20 mm); ^3^ α: degree of fibrin cross-linking (angle tangent to the curve at *K*); ^4^ MA: amplitude of maximum clot strength (total amplitude at the maximum).

## Data Availability

Not applicable.

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
