# Peer review of "Bioabsorbable Carboxymethyl Starch–Calcium Ionic Assembly Powder as a Hemostatic Agent"

_polymers, 2022, doi:10.3390/polym14183909_

Round 1

Reviewer 1 Report

In this manuscript, the authors prepared an ionic assembly of carboxymethyl starch and calcium, which showed elevated hemostatic performance, rheology, adhesion force, C3a complement activation and quick in vivo biodegradation. The paper was well organized and I have the following questions.

1.     Fig. 7. The number of samples used in the C3a complement system activation assay should be disclosed and the error bars should be shown.

2.     Fig. 9. (1) Why did the authors compare the in vivo biodegradation of OOZFIX and Arista AH? (2) Did authors evaluate the biosafety (e.g., tissue toxicity) of hemostatic powder on rats?

Reviewer 2 Report

The scientific paper "Bioabsorbable carboxymethyl starch–calcium ionic assembly powder as a hemostatic agent” aimed to investigate the hemostatic performance of a bioabsorbable powder with decreased desorption. I can make the following considerations:

1)      in 2.8 add ethical approval protocol number and approval date

2)      Section 3 (Results) should be modified for results and discussion. Or the discussion should be done in another section. There is a low number of references (only 18). Therefore, authors need to compare their findings with previous literature, increasing the scientific basis of their research with more references.

3)      Insert study limitations

4)      Remove numerical data from the conclusion.

Round 2

Reviewer 2 Report

No comments